# Cell-to-cell infection by HIV contributes over half of virus infection

**Shingo Iwami**[1,2,3][*][†], **Junko S Takeuchi**[4][†], **Shinji Nakaoka**[5], **Fabrizio Mammano**[6,7], **François Clavel**[6,7], **Hisashi Inaba**[8], **Tomoko Kobayashi**[9], **Naoko Misawa**[4], **Kazuyuki Aihara**[10,11], **Yoshio Koyanagi**[4], **Kei Sato**[3,4][*]

[1]Mathematical Biology Laboratory, Department of Biology, Faculty of Sciences, Kyushu University, Fukuoka, Japan; [2]PRESTO, Japan Science and Technology Agency, Saitama, Japan; [3]CREST, Japan Science and Technology Agency, Saitama, Japan; [4]Laboratory of Viral Pathogenesis, Institute for Virus Research, Kyoto University, Kyoto, Japan; [5]Graduate School of Medicine, University of Tokyo, Tokyo, Japan; [6]INSERM-Genetics and Ecology of viruses, Hospital Saint Louis, Paris, France; [7]Université Paris Diderot, Sorbonne Paris Cité, Paris, France; [8]Graduate School of Mathematical Sciences, University of Tokyo, Tokyo, Japan; [9]Laboratory for Animal Health, Department of Animal Science, Faculty of Agriculture, Tokyo University of Agriculture, Kanagawa, Japan; [10]Institute of Industrial Science, University of Tokyo, Tokyo, Japan; [11]Graduate School of Information Science and Technology, University of Tokyo, Tokyo, Japan

**\*For correspondence:** siwami@
kyushu-u.org (SI); ksato@virus.
kyoto-u.ac.jp (KS)

[†]These authors contributed
equally to this work

**Competing interests:** The
authors declare that no
competing interests exist.

**Reviewing editor**: Arup K
Chakraborty, Massachusetts
Institute of Technology, United
States

**Abstract** Cell-to-cell viral infection, in which viruses spread through contact of infected cell with surrounding uninfected cells, has been considered as a critical mode of virus infection. However, since it is technically difficult to experimentally discriminate the two modes of viral infection, namely cell-free infection and cell-to-cell infection, the quantitative information that underlies cell-to-cell infection has yet to be elucidated, and its impact on virus spread remains unclear. To address this fundamental question in virology, we quantitatively analyzed the dynamics of cell-to-cell and cell-free human immunodeficiency virus type 1 (HIV-1) infections through experimental-mathematical investigation. Our analyses demonstrated that the cell-to-cell infection mode accounts for approximately 60% of viral infection, and this infection mode shortens the generation time of viruses by 0.9 times and increases the viral fitness by 3.9 times. Our results suggest that even a complete block of the cell-free infection would provide only a limited impact on HIV-1 spread.

## Introduction

In in vitro cell cultures and in infected individuals, viruses may display two types of replication strategies: cell-free infection and cell-to-cell infection (*Sattentau, 2008*; *Martin and Sattentau, 2009*; *Talbert-Slagle et al., 2014*). Both transmission means require the assembly of infectious virus particles (*Monel et al., 2012*), which are released in the extracellular medium for cell-free transmission, or concentrated in the confined space of cell-to-cell contacts between an infected cell and bystander target cells in the case of cell-to-cell transmission. It has been shown that most enveloped virus species, including human immunodeficiency virus type 1 (HIV-1), a causative agent of AIDS, spread via cell-to-cell infection, and it is considered that the replication efficacy of cell-to-cell infection is much higher than that of cell-free infection (*Sattentau, 2008*; *Martin and Sattentau, 2009*; *Talbert-Slagle et al., 2014*). However, it is technically impossible to let viruses execute only cell-to-cell infection. In addition, since these two infection processes occur in a synergistic (i.e., nonlinear) manner, the

**eLife digest** Viruses such as HIV-1 replicate by invading and hijacking cells, forcing the cells to make new copies of the virus. These copies then leave the cell and continue the infection by invading and hijacking new cells. There are two ways that viruses may move between cells, which are known as 'cell-free' and 'cell-to-cell' infection. In cell-free infection, the virus is released into the fluid that surrounds cells and moves from there into the next cell. In cell-to-cell infection the virus instead moves directly between cells across regions where the two cells make contact.

Previous research has suggested that cell-to-cell infection is important for the spread of HIV-1. However, it is not known how much the virus relies on this process, as it is technically challenging to perform experiments that prevent cell-free infection without also stopping cell-to-cell infection.

Iwami, Takeuchi et al. have overcome this problem by combining experiments on laboratory-grown cells with a mathematical model that describes how the different infection methods affect the spread of HIV-1. This revealed that the viruses spread using cell-to-cell infection about 60% of the time, which agrees with results previously found by another group of researchers. Iwami, Takeuchi et al. also found that cell-to-cell infection increases how quickly viruses can infect new cells and replicate inside them, and improves the fitness of the viruses.

The environment around cells in humans and other animals is different to that found around laboratory-grown cells, and so more research will be needed to check whether this difference affects which method of infection the virus uses. If the virus does spread in a similar way in the body, then blocking the cell-free method of infection would not greatly affect how well HIV-1 is able to infect new cells. It may instead be more effective to develop HIV treatments that prevent cell-to-cell infection by the virus.

additive (i.e., linear) idea that 'total infection' minus 'cell-free infection' is equal to 'cell-to-cell infection' does not hold true universally. Hence, it was difficult to estimate and compare the efficacies of cell-free and cell-to-cell infection, and different reports provided different estimates (*Dimitrov et al., 1993*; *Carr et al., 1999*; *Chen et al., 2007*; *Sourisseau et al., 2007*; *Zhong et al., 2013*). Thus, the quantitative information that underlies cell-to-cell infection has yet to be elucidated and its impact on virus spread remains unclear.

In this study, through coupled experimental and mathematical investigation, we demonstrate that the efficacy of cell-to-cell HIV-1 infection is 1.4-fold higher than that of cell-free infection (i.e., cell-to-cell infection accounts for approximately 60% of total infection). We also show that the cell-to-cell infection shortens the generation time of viruses by 0.9 times, and increases the viral fitness by 3.9 times. These findings strongly suggest that the cell-to-cell infection plays a critical role in efficient and rapid spread of viral infection. Furthermore, we discuss the role of the cell-to-cell infection in HIV-1 infected individuals, based on in silico simulation with our estimated parameters.

## Results

### Adaptation of a mathematical model to explicitly consider cell-free and cell-to-cell infection

A static cell culture system (i.e., a conventional cell culture system) allows viruses to perform both cell-free and cell-to-cell infection. On the other hand, Sourisseau et al. have reported that the cell-to-cell infection can be prevented by mildly shaking the cell culture infected with viruses (*Sourisseau et al., 2007*). Consistent with the previous report (*Sourisseau et al., 2007*), we verified that shaking did not induce nonspecific consequences on HIV-1 infection (*Figure 2—figure supplement 1*). To quantitatively estimate the efficacy of the cell-free infection and that of the cell-to-cell infection respectively, we adopted this experimental method (see 'Materials and methods'). Static cultures of Jurkat cells, an HIV-1-susceptible human CD4$^+$ T-cell line, allow HIV-1 to propagate both by the cell-free and cell-to-cell infection, while under shaking conditions, Jurkat cells allows HIV-1 to replicate only by the cell-free infection (*Figure 1A*).

Previous mathematical models, which have been widely used for data analyses, essentially describe only the cell-free infection (*Nowak and May, 2000*; *Perelson, 2002*; *Iwami et al., 2012a*, *2012b*) or

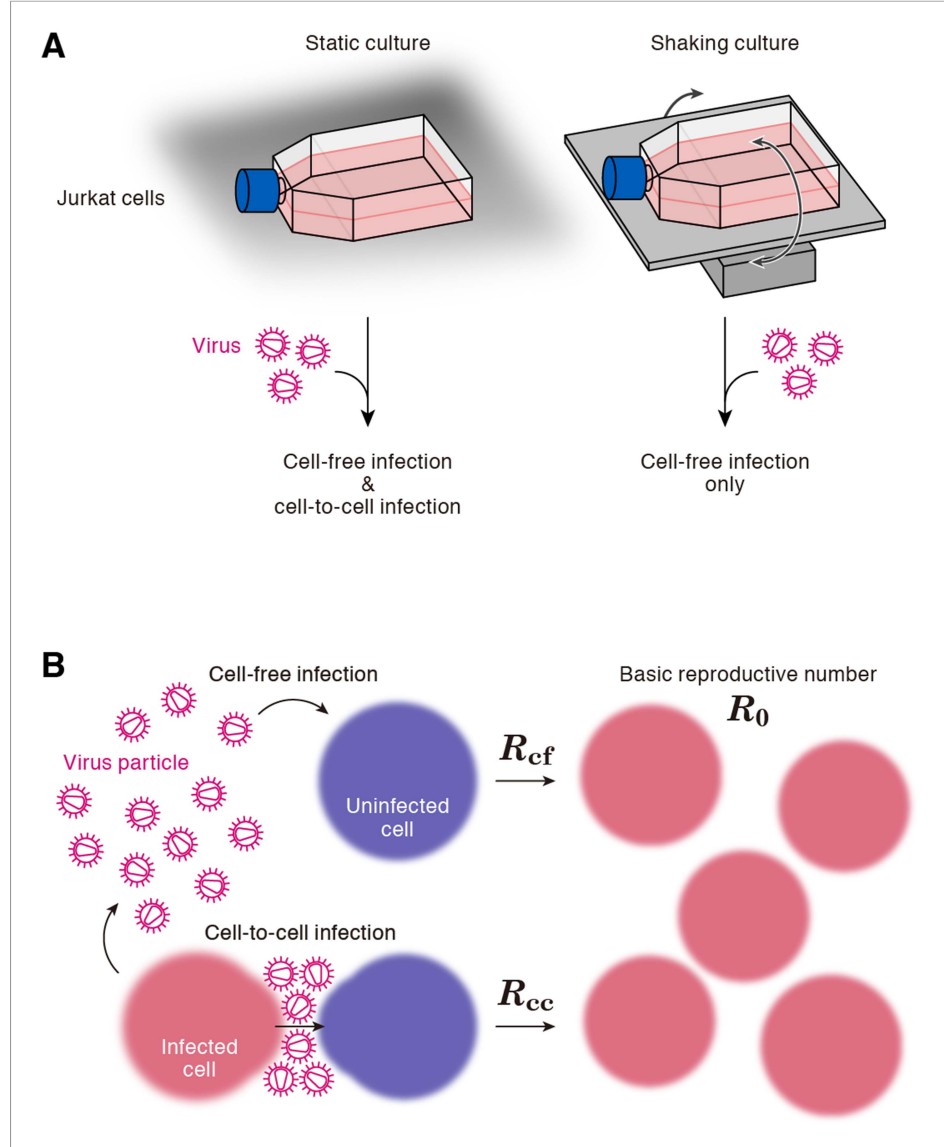

**Figure 1**. Cell culture systems and the basic reproduction number under cell-to-cell and cell-free infection. (**A**) Static and shaking cultures of Jurkat cells. The static and shaking cell cultures allow human immunodeficiency virus type 1 (HIV-1) to perform both cell-free and cell-to-cell infection, and only cell-free infection, respectively. (**B**) The basic reproduction number, $R_0$, is defined as the number of the secondly infected cells produced from a typical infected cell during its infectious period. In the presence of the cell-to-cell and cell-free infection, the basic reproduction number consists of two sub-reproduction numbers through the cell-free infection, $R_{cf}$, and through the cell-to-cell infection, $R_{cc}$, respectively.

implicitly both infection modes (*Komarova and Wodarz, 2013*; *Komarova et al., 2013a, 2013b*). Here we used the following revised model including both infection modes explicitly:

$$\frac{dT(t)}{dt} = gT(t)\left(1 - \frac{T(t) + I(t)}{T_{max}}\right) - \beta T(t)V(t) - \omega T(t)I(t), \tag{1}$$

$$\frac{dI(t)}{dt} = \beta T(t)V(t) + \omega T(t)I(t) - \delta I(t), \tag{2}$$

$$\frac{dV(t)}{dt} = pI(t) - cV(t), \qquad (3)$$

where $T(t)$ and $I(t)$ are the numbers of uninfected and infected cells per ml of a culture, respectively, and $V(t)$ is the viral load measured by the amount of HIV-1 p24 per ml of culture supernatant. The target cells (we used Jurkat cells) grow at a rate $g$ with the carrying capacity of $T_{max}$ (the maximum number of cells in the cell culture flask). The parameters $\beta$, $\delta$, $p$ and $c$ represent the cell-free infection rate, the death rate of infected cells, the virus production rate, and the clearance rate of virions, respectively. Note that $c$, $g$, and $\delta$ include the removal of virus, and of the uninfected and infected cells, due to the experimental samplings. In our earlier works (*Iwami et al., 2012a*, *2012b*; *Fukuhara et al., 2013*; *Kakizoe et al., 2015*), we have shown that the approximating punctual removal as a continuous exponential decay has minimal impact on the model parameters and provides an appropriate fit to the experimental data. In addition, we introduce the parameter $\omega$, describing the infection rate via cell-to-cell contacts (*Sourisseau et al., 2007*; *Sattentau, 2008*; *Sigal et al., 2011*). In the shaking cell culture system, we fixed $\omega = 0$ because the shaking inhibits the formation of cell-to-cell contacts completely (*Sourisseau et al., 2007*). In previous reports, Komarova et al. used a quasi-equilibrium approximation for the number of free virus, and incorporated the dynamics of $V(t)$ into that of $I(t)$ in *Komarova and Wodarz (2013)*, *Komarova et al. (2013a)*, and *Komarova et al. (2013b)*. However, in cell culture system, the clearance of virions usually is not much larger than the death rate of infected cells, like in vivo (see below). This fact does not validate the quasi-equilibrium approximation, and it may affect the quantification of the dynamics of the cell-to-cell and cell-free infection. We introduced the above full model, relying on a carefully designed experiment, to accurately extract the quantitative information that underlies HIV-1 infection. Furthermore, our experimental datasets include all time-series of the number of uninfected, infected cell, and virions. Thus, our coupled experimental and mathematical investigations with a sufficient datasets allowed us to estimate all parameters in *Equations 1–3*, and to compute the basic reproduction number, generation time, and Malthus coefficient (see below).

## Data fitting to quantify the cell-free and cell-to-cell contribution to HIV spread

Correctly estimated parameter sets with possible variation are required to reproduce model prediction for pure cell-to-cell infection in silico. However, point estimation of the model parameter set by a conventional ordinary least square method does not capture possible variations of kinetic parameters and model prediction. To assess the variability of kinetic parameters and model prediction, we perform Bayesian estimation for the whole dataset using Markov Chain Monte Carlo (MCMC) sampling (see 'Materials and methods' and *Supplementary file 1*), and simultaneously fit *Equations 1–3* with $\omega > 0$ and $\omega = 0$ to the concentration of p24-negative and -positive Jurkat cells and the amount of p24 viral protein in the static and shaking cell cultures, respectively. Here we note that $g$ and $T_{max}$ were separately estimated and fixed to be $0.47 \pm 0.10$ for the static culture and $0.54 \pm 0.09$ for the shaking culture per day, and $(1.51 \pm 0.02) \times 10^6$ and $(1.22 \pm 0.02) \times 10^6$ cells per flask of medium from the cell growth experiments, respectively (see 'Materials and methods', *Figure 2—figure supplement 2* and *Supplementary file 2*). In addition, we used $c$ value of 2.3 per day, which is estimated from daily harvesting of viruses (i.e., the amount of p24 have to be reduced by around 90% per day by the daily medium-replacement procedure).

The remaining four common parameters $\beta$, $\omega$, $\delta$ and $p$, along with the six initial values for $T(0)$, $I(0)$ and $V(0)$ in the static and the shaking cell cultures, were determined by fitting the model to the data. Experimental measurements, which were below the detection limit, were excluded in the fitting. The estimated parameters of the model and derived quantities are given in *Table 1*, and the estimated initial values are summarized in *Supplementary file 3*. The typical behavior of the model using these best-fit parameter estimates is shown together with the data in *Figure 2*, which reveals that *Equations 1–3* describe these in vitro data very well. The shadowed regions correspond to 95% posterior predictive intervals, the dashed lines give the best-fit solution (mean) for *Equations 1–3*, and the dots show the experimental datasets. This suggests that the parameters that were estimated are representative for the various processes underlying the HIV-1 kinetics including the cell-to-cell and cell-free infection.

**Table 1.** Parameters estimated by mathematical-experimental analysis

| Parameter name | Symbol | Unit | Exp. 1 | Exp. 2 | Exp. 3 | Ave. ± S.D. |
|---|---|---|---|---|---|---|
| Parameters obtained from simultaneous fit to time-course experimental dataset | | | | | | |
| Rate constant for cell-free infection | $\beta$ | $10^{-6} \times$ (p24 day)$^{-1}$ | 5.59* (3.54–8.41)† | 3.27 (2.05–5.01) | ‡3.70 (2.28–5.77) | 4.18 ± 1.41 |
| Rate constant for cell-to-cell infection | $\omega$ | $10^{-6} \times$ (cell day)$^{-1}$ | 0.88 (0.45–1.39) | 1.25 (0.70–1.97) | 1.13 (0.64–1.79) | 1.09 ± 0.33 |
| Production rate of total viral protein | $p$ | day$^{-1}$ | 0.37 (0.22–0.59) | 0.59 (0.34–0.92) | 0.54 (0.31–0.86) | 0.50 ± 0.16 |
| Death rate of infected cells | $\delta$ | day$^{-1}$ | 0.45 (0.32–0.64) | 0.54 (0.38–0.75) | 0.50 (0.36–0.68) | 0.50 ± 0.10 |
| Quantities derived from fitted values | | | | | | |
| Basic reproduction number through cell-free infection | $R_{cf}$ | – | 2.88 (2.34–3.53) | 2.27 (1.98–2.66) | 2.43 (2.04–2.95) | 2.44 ± 0.23 |
| Basic reproduction number through cell-to-cell infection | $R_{cc}$ | – | 2.95 (1.48–4.70) | 3.65 (1.77–6.05) | 3.39 (1.82–5.38) | 3.39 ± 0.91 |
| Basic reproduction number | $R_0$ | – | 5.83 (4.20–7.75) | 5.92 (3.99–8.46) | 5.83 (4.21–7.89) | 5.83 ± 0.94 |
| Contribution of cell-to-cell infection | $\frac{R_{cc}}{R_{cf}+R_{cc}}$ | – | 0.50 (0.34–0.63) | 0.60 (0.44–0.72) | 0.57 (0.43–0.70) | 0.57 ± 0.07 |

*Mean value.
†95% confidence interval.
‡Average and standard deviation of merged values in experiment 1, 2, and 3.

Our model (i.e., *Equations 1–3*) applied to time-course experimental data under static and shaking conditions (i.e., *Figure 2A* and *Figure 2B*, respectively) allowed to extract the kinetic parameters in the model (see *Table 1*), in particular the rate constant for the cell-free infection ($\beta$) and the rate constant for the cell-to-cell infection ($\omega$). However, from the estimated values of $\beta$ and $\omega$, we could not directly compare the efficiency of the two infection modes, because of the different units of measure of these parameters (p24/day for $\beta$, and cells/day for $\omega$). To quantify each infection mode and overcome the above difficulty, we derived the basic reproduction number $R_0$ (*Perelson and Nelson, 1999*; *Nowak and May, 2000*; *Iwami et al., 2012b*), an index reflecting the average number of newly infected cells produced from any one infected cell (see mathematical appendix in 'Materials and methods'). Note that secondly infected cells are produced from both the cell-free and cell-to-cell infection. Interestingly, in spite of nonlinear interaction between the two modes of virus transmission, our derivation of $R_0$ revealed that the secondly infected cells were the sum of the basic reproduction number through the cell-free infection $R_{cf} = \beta p T_{max}/\delta c$ and the basic reproduction number through the cell-to-cell infection $R_{cc} = \omega T_{max}/\delta$, (i.e., $R_0 = R_{cf} + R_{cc}$) (see *Figure 1B*). Using all accepted MCMC parameter estimates from the time-course experimental datasets, we calculated that on average the mean of the total basic reproductive number is $R_0 = 5.83 \pm 0.94$ (average ± standard deviation), and the mean number of secondly infected cells through the cell-free infection and the cell-to-cell infection are $R_{cf} = 2.44 \pm 0.23$ and $R_{cc} = 3.39 \pm 0.91$, respectively (see *Table 1*). The distributions of calculated $R_0$, $R_{cf}$, and $R_{cc}$, are shown in *Figure 3A–C*, respectively. These estimates indicate that the contribution of the cell-to-cell infection is almost 60% on average (i.e., $R_{cc}/(R_{cc} + R_{cf}) = 0.57 \pm 0.07$: *Table 1*) and this mode of infection is predominant during the HIV-1 spread in Jurkat cells. In *Figure 3D*, the distributions of calculated ratio are shown. Interestingly, this estimation is consistent with that by *Komarova and Wodarz (2013)*, *Komarova et al. (2013a)*, and *Komarova et al. (2013b)*, although they did not take into account the difference of the death rate in the shaking and static conditions.

## Advantage of cell-to-cell infection

We also derived the viral generation time, defined as the time it takes for a population of virions to infect cells and reproduce (*Perelson and Nelson, 1999*), from *Equations 1–3* in the static and shaking cell cultures (see mathematical appendix in 'Materials and methods'). In the presence and absence of the cell-to-cell infection (i.e., for the static and shaking cell cultures, respectively), the mean generation time is calculated as $1/\delta + R_{cf}/cR_0 = 2.22 \pm 0.32$ days and $1/\delta + 1/c = 2.47 \pm 0.32$ days, respectively (see *Table 2*). Thus, cell-to-cell infection shortens the generation time by on average 0.90

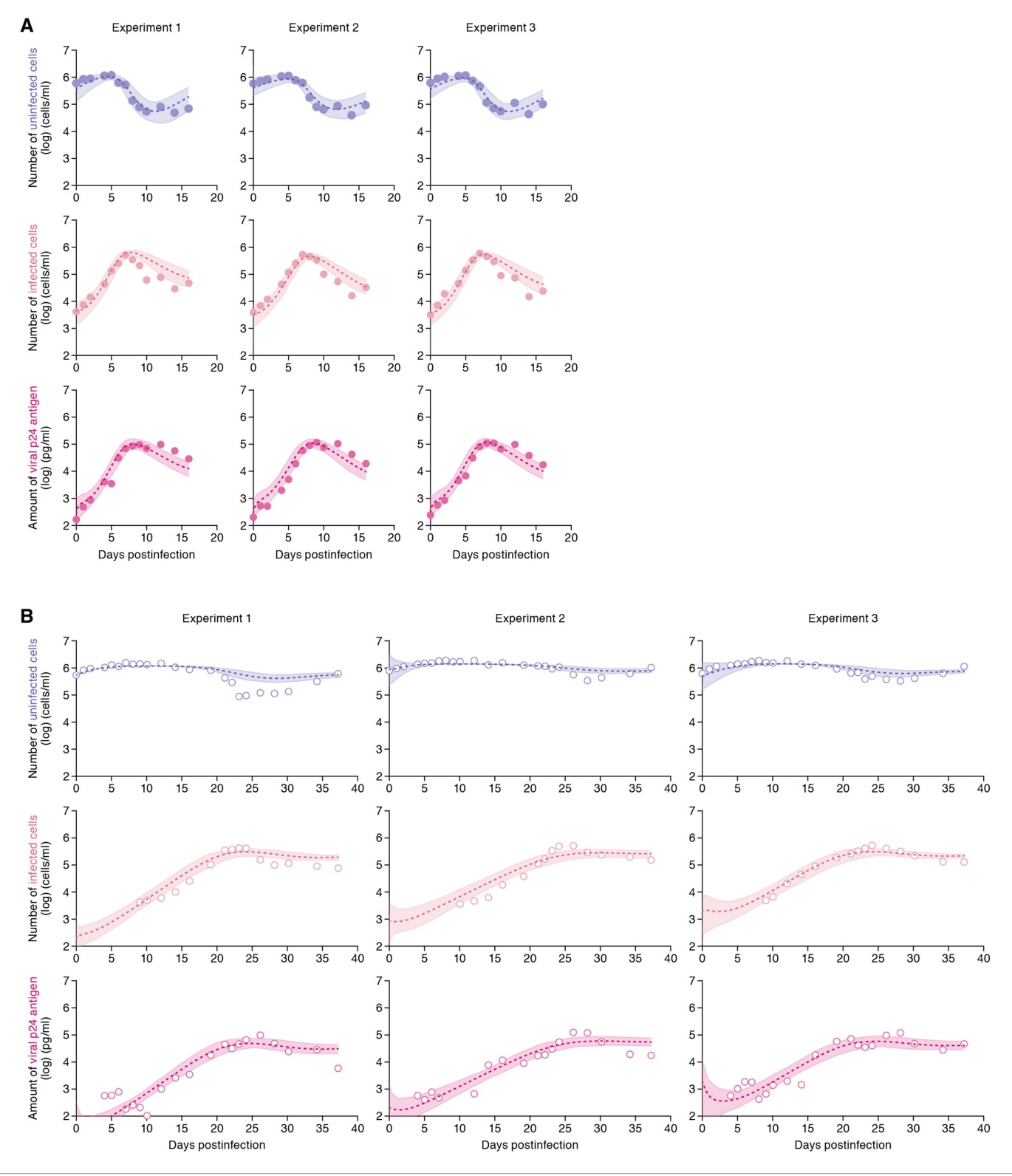

**Figure 2**. Dynamics of HIV-1 infection in Jurkat cells through cell-free and cell-to-cell infection. Jurkat cells were inoculated with HIV-1 (at multiplicity of infection 0.1) in the static and shaking cell cultures. Panels **A** and **B** show the time-course of experimental data for the numbers of uninfected cells (top) and infected cells (middle), and the amount of viral protein p24 (bottom) in the static and shaking cell culture systems, respectively. The shadow regions
*Figure 2. continued on next page*

*Figure 2. Continued*

correspond to 95% posterior predictive intervals, the dashed curves give the best-fit solution (mean) for *Equations 1–3* to the time-course dataset. All data in each experiment were fitted simultaneously. In panels **A** and **B**, the results of three independent experiments are respectively shown.

The following figure supplements are available for figure 2:

**Figure supplement 1**. No effect of the shaking procedure on HIV-1 cell-free infection.

**Figure supplement 2**. Dynamics of Jurkat cell growth.

**Figure supplement 3**. Dot plots of infected cells by flow cytometry.

times, and enables HIV-1 to efficiently infect target cells (*Sato et al., 1992*; *Carr et al., 1999*). Furthermore, we calculated the Malthus coefficient, defined as the fitness of virus (*Nowak and May, 2000*; *Nowak, 2006*) (or the speed of virus infection) (see mathematical appendix in 'Materials and methods'). In the presence and absence of the cell-to-cell infection, the Malthus coefficient is calculated as 1.86 ± 0.37 and 0.49 ± 0.05 per day, respectively (see *Table 2*). Thus, cell-to-cell infection increases the HIV-1 fitness by 3.80-fold (corresponding to 944-fold higher viral load 5 days after the infection) and plays an important role in the rapid spread of HIV-1. Thus, the efficient viral spread via the cell-to-cell infection is relevant, especially at the beginning of virus infection.

## Virtual experiments of cell-to-cell infection in silico

While the shaking culture prevents the cell-to-cell infection, it is technically difficult to completely block the cell-free infection. Here, using our estimated kinetic parameters (*Table 1* and *Supplementary file 3*), we carried out a 'virtual experiment' eliminating the contribution of the cell-free infection using all accepted MCMC estimated parameter values, allowing to estimate only the cell-to-cell infection, in silico (see *Figure 4*). Our simulated mean values (represented by solid lines) of the cell-to-cell infection of HIV-1 are consistently located between the time course of experimental data under the static conditions (closed circles, including both the cell-free and cell to cell infections) and those under the shaking conditions (open circles, reflecting only the cell-free infection). The shadowed regions correspond to 95% posterior predictive intervals. In terms of the dynamics of infected cells and virus production, the simulated values corresponding to cell-to-cell virus propagation, are closer to experimental data from the coupled cell-free and cell-to-cell infection, than to data from the cell-free infection only. This shows that the cell-free infection, which contributes approximately 40% to the whole HIV-1 infection process, plays a limited role on the virus spread. In other words, even if we could completely block the cell-free infection, the cell-to-cell infection would still effectively spread viruses (*Sigal et al., 2011*). We address this point in 'Discussion'.

## Discussion

Through experimental-mathematical investigation, here we quantitatively elucidated the dynamics of the cell-to-cell and cell-free HIV-1 infection modes (*Figure 2* and *Table 1*). We derived the basic reproduction number, $R_0$, and divided it into the numbers of secondly infected cells through the cell-free infection, $R_{cf}$, and the cell-to-cell infection, $R_{cc}$, respectively (*Figure 1B* and mathematical appendix in 'Materials and methods'). Based on our calculated values of these three indexes, we found that about 60% of the viral infection is attributed to the cell-to-cell infection in the in vitro cell culture system (*Table 1*), which is consistent with previous estimation by *Komarova and Wodarz (2013)*, *Komarova et al. (2013a)*, and *Komarova et al. (2013b)*. In addition, we revealed that the cell-to-cell infection effectively promotes the virus infection by reducing the generation time (×0.9 times), and by increasing the Malthus coefficient (×3.80 times) (*Table 2*).

When we consider the significance of the cell-to-cell infection in patients infected with HIV-1, it should be noted that the environment of immune cells including CD4+ T-cells in vivo is radically different from the conditions of in vitro cell cultures. For instance, lymphocytes are closely packed in lymphoid tissues such as lymph nodes, and thereby, the frequency for the infected cell to contact with adjacent uninfected cells in vivo would be much higher than that in in vitro cell cultures. In addition, Murooka et al.

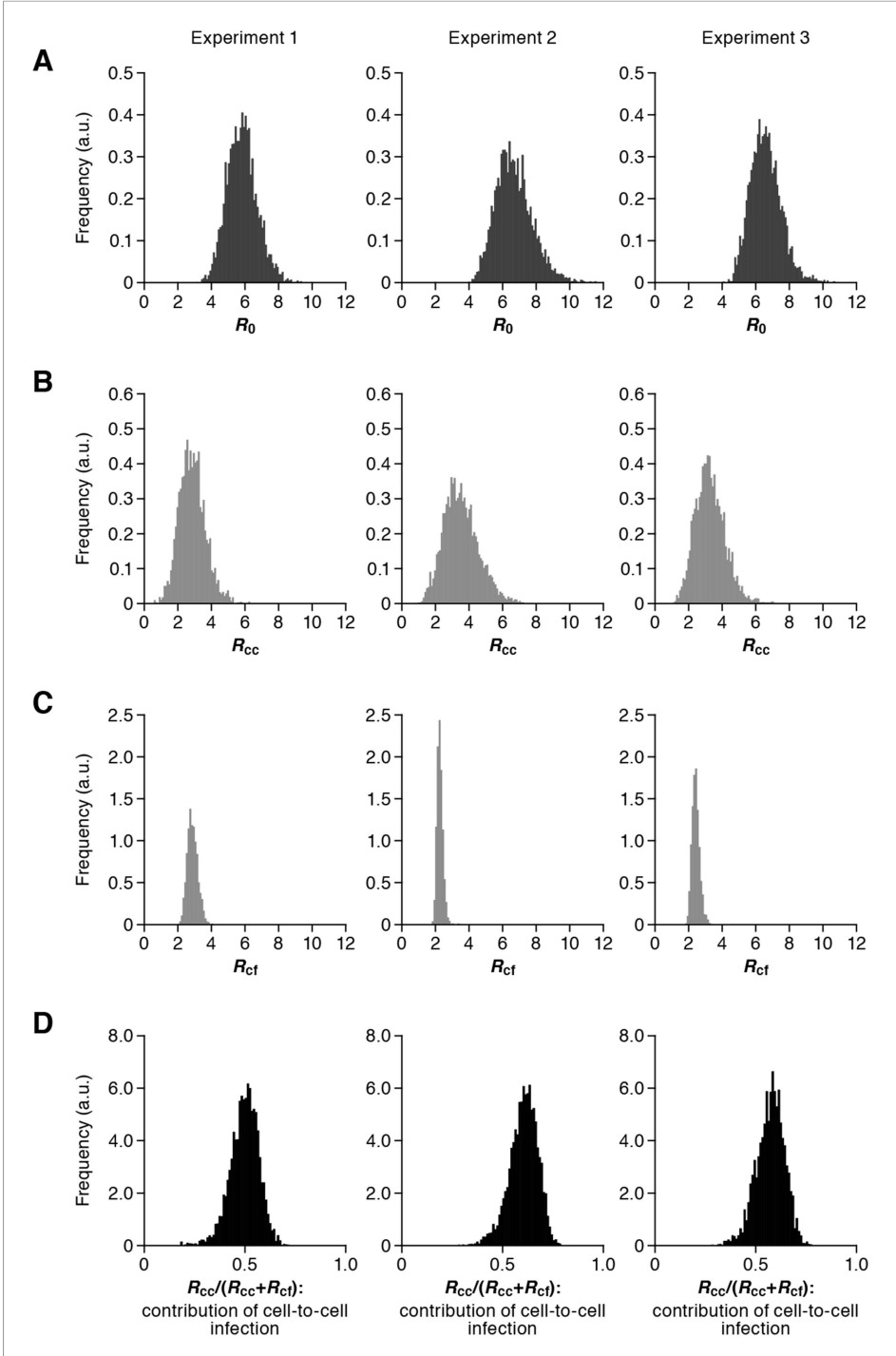

**Figure 3.** Distribution of the basic reproduction numbers, generation time, and Malthus coefficient. The distribution of the basic reproduction number, $R_0$, the number of secondary infected cells through the cell-free infection, $R_{cf}$, and the cell-to-cell infection, $R_{cc}$, calculated from all accepted Markov Chain Monte Carlo (MCMC) parameter estimates are shown in **A**, **B**, and **C**, respectively. The contribution of the cell-to-cell infection (i.e., $R_{cc}/(R_{cf} + R_{cc})$) is distributed as in **D**. For each plot, the last 15,000 MCMC samples among the total 50,000 samples are used. a.u., arbitrary unit.

have directly demonstrated that HIV-1-infected cells converge to lymph nodes and can be vehicles for viral dissemination in vivo (**Murooka et al., 2012**). Moreover, certain studies have suggested that cell-to-cell viral spread is resistant to anti-viral immunity such as neutralizing antibodies and

**Table 2**. Generation time and Malthus coefficient of virus infection

| Cell culture system | Exp. 1 | Exp. 2 | Exp. 3 | Ave. ± S.D. |
|---|---|---|---|---|
| Generation time of HIV-1 | | | | |
| Static cell culture | 2.51* days | 2.08 days | 2.22 days | (2.22 ± 0.32)‡ days |
| | (1.78–3.38) days | (1.54–2.78) days | (1.69–2.93) days | – |
| Shaking cell culture | 2.73† days | 2.34 days | 2.47 days | (2.47 ± 0.32) days |
| | (1.99–3.59) days | (1.77–3.06) days | (1.91–3.18) days | – |
| Malthus coefficient of HIV-1 | | | | |
| Static cell culture | 1.61 day$^{-1}$ | 2.03 day$^{-1}$ | 1.86 day$^{-1}$ | (1.86 ± 0.37) day$^{-1}$ |
| | (1.10–2.27) day$^{-1}$ | (1.32–3.01) day$^{-1}$ | (1.26–2.72) day$^{-1}$ | – |
| Shaking cell culture | 0.57 day$^{-1}$ | 0.46 day$^{-11}$ | 0.49 day$^{-1}$ | (0.49 ± 0.05) day$^{-1}$ |
| | (0.47–0.67) day$^{-1}$ | (0.38–0.56) day$^{-1}$ | (0.39–0.61) day$^{-1}$ | – |

*Mean value.
†95% confidence interval.
‡Average and standard deviation of merged values in experiment 1, 2, and 3.
HIV-1, human immunodeficiency virus type 1.

cytotoxic T lymphocytes (*Martin and Sattentau, 2009*). Therefore, these notions strongly suggest that the contribution of the cell-to-cell infection for viral propagation in vivo may be much higher than that estimated from the in vitro cell culture system.

As another significance of cell-to-cell viral spread, Sigal et al. have suggested that the cell-to-cell infection permits viral replication even under the anti-retroviral therapy (*Sigal et al., 2011*). This is attributed to the fact that the multiplicity of infection per cell is tremendously higher than that reached by an infectious viral particle. However, in the previous report (*Sigal et al., 2011*), the contribution of the cell-to-cell infection remained unclear. To further understand the role of the cell-to-cell infection, we quantified the contributions of the cell-to-cell and cell-free infection modes (*Table 1*). Interestingly, we found that the cell-to-cell infection mode is predominant during the infection. Furthermore, our virtual experiments showed that a complete block of the cell-free infection, which is highly susceptible to current antiviral drugs, provides only a limited impact on the whole HIV-1 infection (*Figure 3*). Taken together, our findings further support that the cell-to-cell infection can be a barrier to prevent the cure of HIV-1 infection, which is discussed in *Sigal et al. (2011)*. However, it should be noted that some papers have shown that cell-to-cell spread cannot overcome the action of most anti-HIV-1 drugs (*Titanji et al., 2013*; *Agosto et al., 2014*). To fully elucidate this issue, further investigations will be needed.

In addition to HIV-1, other viruses such as herpes simplex virus, measles virus, and human hepatitis C virus drive their dissemination via cell-to-cell infection (*Sattentau, 2008*; *Talbert-Slagle et al., 2014*). Although the impact of cell-to-cell viral spread is a topic of broad interest in virology, it was difficult to explore this issue by conventional virological experiments, because an infected cell is simultaneously capable of achieving cell-to-cell infection along with producing infectious viral particles. By applying mathematical modeling to the experimental data, here we estimated the sole dynamics of cell-free infection in the cell culture system. The synergistic strategy of experiments with mathematical modeling is a powerful approach to quantitatively elucidate the dynamics of virus infection in a way that is inaccessible through conventional experimental approaches.

## Materials and methods

### Cell culture and HIV-1 infection

Jurkat cell line (*Watanabe et al., 2012*) was cultured in the culture medium: RPMI 1640 (Sigma, St. Louis, MO) containing 2% fetal calf serum and antibiotics. The virus solution was prepared as previously described (*Sato et al., 2010*, *2013*, *2014*; *Iwami et al., 2012a*). Briefly, 30 µg of pNL4-3

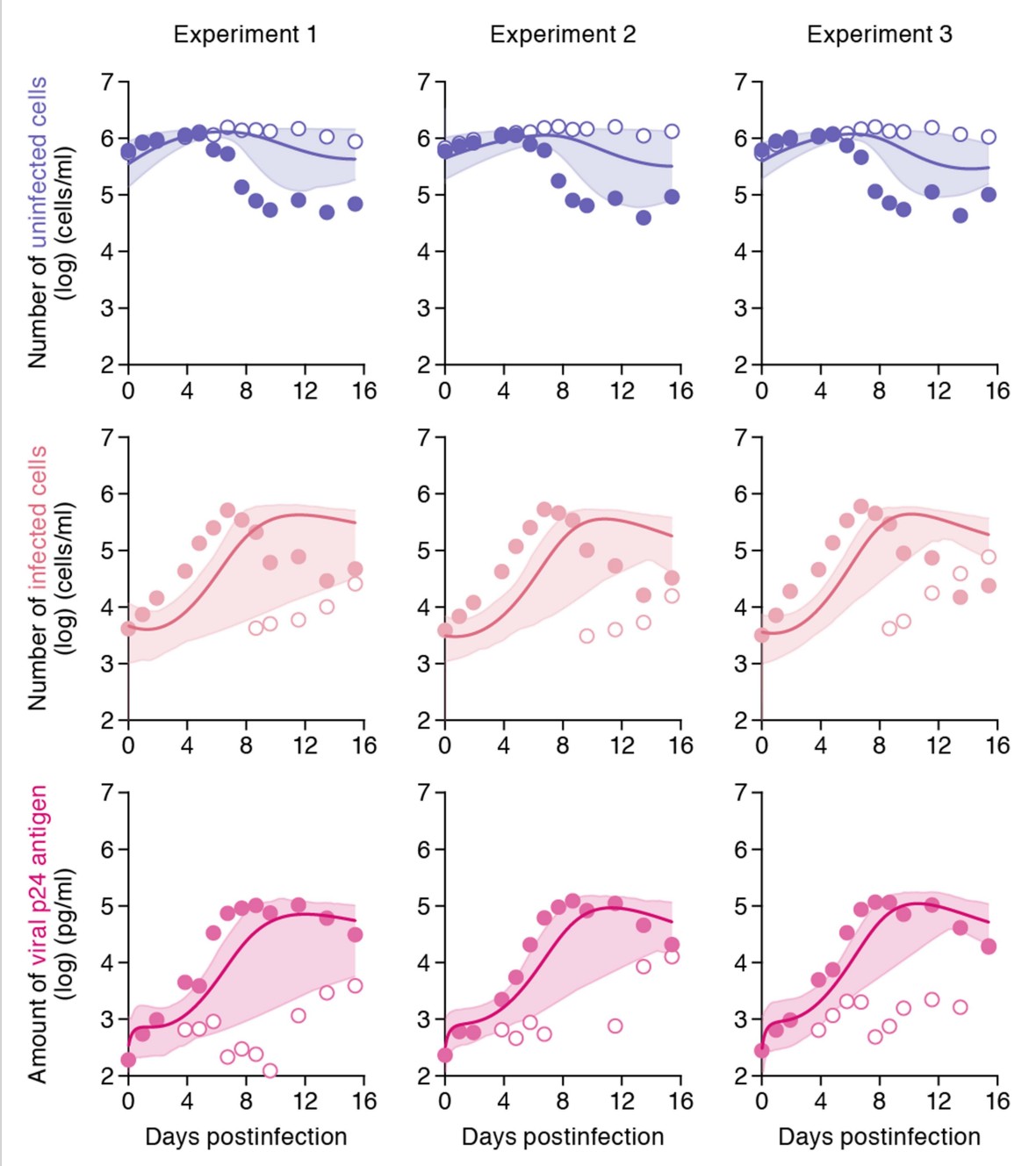

**Figure 4**. Simulating cell-to-cell infection of HIV-1. Using our estimated parameters, the pure cell-to-cell infection is simulated in silico (solid curves). The simulated values are located between the time course of experimental data under the static conditions (closed circles) and those under the shaking conditions (open circles). The shadowed regions correspond to 95% posterior predictive intervals.

plasmid (**Adachi et al., 1986**) (GenBank accession no. M19921.2) was transfected into 293T cells by the calcium-phosphate method. At 48 hr post-transfection, the culture supernatant was harvested, centrifuged, and then filtered through a 0.45-μm-pore-size filter to produce virus solution. The infectivity of virus solution was titrated as previously described (**Iwami et al., 2012a**). Briefly, the virus solution obtained was serially diluted and then inoculated onto phytohemagglutinin-stimulated human peripheral blood mononuclear cells in a 96-well plate in triplicate. At 14 days postinfection, the endpoint was determined by using an HIV-1 p24 antigen enzyme-linked immunosorbent assay

(ELISA) kit (ZetptoMetrix, Buffalo, NY) according to the manufacture's procedure, and virus infectivity was calculated as the 50% tissue culture infectious doses (TCID$_{50}$) according to the Reed-Muench method.

For HIV-1 infection, $3 \times 10^5$ of Jurkat cells were infected with HIV-1 (multiplicity of infection 0.1) at 37°C for 2 hr. The infected cells were washed three times with the culture medium, and then suspended with 3 ml of culture medium and seeded into a 25-cm$^2$ flask (Nunc, Rochester, NY). For the static infection, the infected cell culture was kept in a 37°C/5% CO$_2$ incubator as usual. For the shaking infection, the infected cell culture was handled as previously described (*Sourisseau et al., 2007*). Briefly, the cell culture was kept on a Petit rocker Model-2230 (Wakenyaku, Japan) placed in 37°C/5% CO$_2$ incubator, and was gently shaken at 40 movements per min. The amount of virus particles in the culture supernatant and the number of infected cells were routinely measured as follows: a portion (300 μl) of the infected cell culture was routinely harvested, and the amount of released virions in the culture supernatant was quantified by using an HIV-1 p24 antigen ELISA kit (ZetptoMetrix) according to the manufacture's procedure. The cell number was counted by using a Scepter handled automated cell counter (Millipore, Germany) according to the manufacture's protocol. The percentage of infected cells was measured by flow cytometry. Flow cytometry was performed with a FACSCalibur (BD Biosciences, San Jose, CA) as previously described (*Sato et al., 2010*; *Sato et al., 2011*, *2013*, *2014*; *Iwami et al., 2012a*), and the obtained data were analyzed with CellQuest software (BD Biosciences). For flow cytometry analysis, a fluorescein isothiocyanate-labeled anti-HIV-1 p24 antibody (KC57; Beckman Coulter, Pasadena, CA) was used. The representative dot plots are shown in *Figure 2—figure supplement 3*. The data is available upon request. The remaining cell culture was centrifuged and then resuspended with 3 ml of fresh culture medium. It should be noted that the procedure for HIV-1 infection was performed at time $t = -2$ day in the figures. Because there is no viral protein production in the first day after infection, each in vitro experimental quantity was measured daily from $t = 0$ day (i.e., 2 days after HIV-1 inoculation). The detection threshold of each value are the followings: cell number (cell counting), 3000 cells/ml; % p24-positive cells (flow cytometry), 0.3%; and p24 antigen in culture supernatant (p24 antigen ELISA), 80 pg/ml.

## Parameter estimation

A statistical model adopted in the Bayesian inference assumes measurement error to follow normal distribution with mean zero and unknown variance (error variance). A distribution of error variance is also inferred with the Gamma distribution as its prior distribution. Posterior predictive parameter distribution as an output of MCMC computation represents parameter variability. Distributions of model parameters and initial values were inferred directly by MCMC computations. On the other hand, distributions of the basic reproduction numbers and the other quantities were calculated from the inferred parameter sets (*Figure 3* for graphical representation). A set of computations for *Equations 1–3* with estimated parameter sets gives a distribution of outputs (virus load and cell density) as model prediction. To investigate variation of model prediction, global sensitivity analyses were performed. The range of possible variation is drawn in *Figure 2* as 95% confidence interval. Technical details of MCMC computations are summarized in *Supplementary file 1*.

## Quantification of Jurkat cell growth

We here estimate the growth kinetics of Jurkat cells, which have been commonly used for HIV-1 studies, under the normal (i.e., mock-infected) condition with the following mathematical model:

$$\frac{dT(t)}{dt} = gT(t)\left(1 - \frac{T(t)}{T_{max}}\right), \tag{4}$$

where the variable $T(t)$ is the number of Jurkat cells at time t and the parameters $g$ and $T_{max}$ are the growth rate of the cells (i.e., Log2/$g$ is the doubling time) and the carrying capacity of the cell culture flask, respectively. Nonlinear least-squares regression (*FindMinimum* package of *Mathematica9.0*) was performed to fit *Equation 4* to the time-course numbers of Jurkat cells in the normal condition. The fitted parameter values are listed in *Supplementary file 2* and the model behavior using these best-fit parameter estimates is presented together with the data in *Figure 2—figure supplement 2*.

## Mathematical appendix

The linearized equation of *Equations 1–3* at the virus-free steady state, $(T_{max}, 0, 0)$, is given as follows:

$$\frac{dI(t)}{dt} = \beta T_{max}V(t) + \omega T_{max}I(t) - \delta I(t), \tag{5}$$

$$\frac{dV(t)}{dt} = pI(t) - cV(t). \tag{6}$$

Let $b(t)$ be the number of newly produced infected cells in the linear phase:

$$b(t) := \beta T_{max}V(t) + \omega T_{max}I(t). \tag{7}$$

Applying the variation of constants formula to *Equations 5, 6*, we have

$$V(t) = V(0)e^{-ct} + \int_0^t e^{-c(t-s)}pI(s)ds, \tag{8}$$

$$I(t) = I(0)e^{-\delta t} + \int_0^t e^{-\delta(t-z)}b(z)dz. \tag{9}$$

Inserting *Equation 9* into *Equation 8* to exchange the order of integrals, we have

$$V(t) = g(t) + p\int_0^t \int_0^x e^{-c(x-\theta)-\delta\theta}d\theta b(t-x)dx, \tag{10}$$

where

$$g(t) := V(0)e^{-ct} + \int_0^t e^{-c(t-s)}pI(0)e^{-\delta t}ds.$$

From *Equation 7* and *Equations 9, 10*, we arrive at the following renewal equation:

$$b(t) = h(t) + \int_0^t \Psi(x)b(t-x)dx,$$

where $h(t)$ is given by

$$h(t) := \omega T_{max}I(0)e^{-\delta t} + \beta T_{max}g(t),$$

and the kernel $\Psi(x)$ is given by

$$\Psi(x) := \beta T_{max}p\int_0^x e^{-\delta\theta-c(x-\theta)}d\theta + \omega T_{max}e^{-\delta x},$$

$$= \frac{\beta T_{max}p}{\delta c}(\phi_1 * \phi_2)(x) + \frac{\omega T_{max}}{\delta}\phi_1(x).$$

In the above expression, $\phi_j(x)$ denotes the probability density function given by

$$\phi_1(x) = \delta e^{-\delta x}, \quad \phi_2(x) = ce^{-cx},$$

and, $*$ denotes the convolution of functions. From the general theory of the basic reproduction number (*Inaba, 2012*), $R_0$ for the reproduction of infected cells is given by

$$R_0 = \int_0^\infty \Psi(x)dx = \frac{\beta T_{max}p}{\delta c} + \frac{\omega T_{max}}{\delta} = R_{cf} + R_{cc},$$

where $R_{cf}$ and $R_{cc}$ denote the reproduction numbers for infected cells mediated by the cell-free and cell-to-cell infection, respectively.

Next we consider the reproduction process of viruses. Let $\rho(t):= pI(t)$ be the number of newly produced viruses at time $t$. From *Equations 8, 9*, we obtain

$$\rho(t) = pI(0)e^{-\delta t} + \int_0^t e^{-\delta(t-z)}(\beta T_{max}pV(z) + \omega T_{max}\rho(z))dz, \qquad (11)$$

where

$$V(t) = V(0)e^{-ct} + \int_0^t e^{-c(t-s)}\rho(s)ds. \qquad (12)$$

Inserting *Equation 11* into *Equation 12*, we again arrive at the following renewal equation:

$$\rho(t) = q(t) + \int_0^t \Psi(x)\rho(t-x)dx,$$

where $q(t)$ is given by

$$q(t):= pI(0)e^{-\delta t} + \int_0^t e^{-\delta(t-z)}p\beta T_{max}V(0)e^{-cz}dz.$$

Note that the reproduction kernel $\Psi(x)$ for the virus reproduction is the same as the kernel for the cell reproduction. Thus the probability density function of the virus reproduction is given by

$$\psi(x):= \frac{\Psi(x)}{R_0} = \frac{R_{cf}}{R_0}(\phi_1 * \phi_2)(x) + \frac{R_{cc}}{R_0}\phi_1(x).$$

Then the generation time for the virus reproduction, denoted by $G$, is calculated as follows:

$$G:= \int_0^\infty t\psi(t)dt = \frac{R_{cf}}{R_0}G_{cf} + \frac{R_{cc}}{R_0}G_{cc} \le G_{cf},$$

where $G_{cf}:= 1/\delta + 1/c$ and $G_{cc}:= 1/\delta$ are the generation times for virus reproduction mediated by the cell-free and cell-to-cell infection, respectively.

The Malthusian coefficient for the virus reproduction must be given as the dominant real root of the Euler-Lotka equation as

$$\int_0^\infty e^{-\lambda x}\Psi(x)dx = \frac{\beta T_{max}p}{\delta c}\widehat{\phi_1}(\lambda)\widehat{\phi_2}(\lambda) + \frac{\omega T_{max}}{\delta}\widehat{\phi_1}(\lambda) = 1,$$

where $\widehat{\phi_j}$ denotes the Laplace transformation of a function $\phi_j$. That is,

$$\widehat{\phi_1}(\lambda) = \int_0^\infty e^{-\lambda x}\phi_1(x)ds = \frac{\delta}{\delta + \lambda}, \quad \widehat{\phi_2}(\lambda) = \int_0^\infty e^{-\lambda x}\phi_2(x)ds = \frac{c}{c + \lambda}.$$

Therefore the Euler-Lotka equation can be calculated explicitly as follows:

$$\frac{\beta T_{max}p}{\delta c}\frac{\delta c}{(\delta + \lambda)(c + \lambda)} + \frac{\omega T_{max}}{\delta}\frac{\delta}{\delta + \lambda} = 1,$$

which is reduced to a quadratic equation,

$$\lambda^2 + \delta c\big(G_{cc} + (1 - R_{cc})(G_{cf} - G_{cc})\big)\lambda + \delta c(1 - R_0) = 0. \qquad (13)$$

If $R_0 > 1$, *Equation 13* has a unique positive root, which is no other than the Malthusian coefficient for the virus reproduction, so it is calculated as,

$$\lambda = \frac{-\delta c\big(G_{cc} + (1 - R_{cc})(G_{cf} - G_{cc})\big) + \sqrt{\delta^2 c^2\big(G_{cc} + (1 - R_{cc})(G_{cf} - G_{cc})\big)^2 - 4\delta c(1 - R_0)}}{2}.$$

## Acknowledgements

This work was supported in part by JST PRESTO program (to SI); JST CREST program (to SI, HI, KA, and KS); Grants-in-Aid for Young Scientists B25800092 (to SI) and B25871132 (to SN) from the Japan Society for the Promotion of Science (JSPS); JSPS KAKENHI Grant Number 10192783 and 15KT0107 (to SI), 25400194 (to HI) and 15K07166 (to KS); Inamori Foundation (to SI); the Aihara Innovative Mathematical Modeling Project, JSPS, through the 'Funding Program for World-Leading Innovative R & D on Science and Technology (FIRST Program)', initiated by Council for Science and Technology Policy (to SI, SN, HI, KA, and KS); the Japan Agency for Medical Research and Development, AMED (H27-ShinkoJitsuyoka-General-016) (to SI, SN, HI, KA, and KS); Agence Nationale de Recherches sur le Sida et les Hepatites Virales (ANRS) (to FM); a Grant-in-Aid for Scientific Research on Innovative Areas 24115008 from the Ministry of Education, Culture, Sports, Science and Technology of Japan (to YK); JSPS Core-to-Core program, A. Advanced Research Networks (to YK); Research on HIV/AIDS from AMED 15Afk0410013h0001 (to YK); Takeda Science Foundation (to KS); Sumitomo Foundation Research Grant (to KS); Senshin Medical Research Foundation (to KS); Imai Memorial Trust for AIDS Research (to KS); Ichiro Kanehara Foundation (to KS); Kanae Foundation for the Promotion of Medical Science (to KS); Suzuken Memorial Foundation (to KS); Uehara Memorial Foundation (to KS).

## Additional information

### Funding

| Funder | Author |
| --- | --- |
| Japan Science and Technology Agency | Shingo Iwami |

The funder had no role in study design, data collection and interpretation, or the decision to submit the work for publication.

### Author contributions

SI, FM, Conception and design, Analysis and interpretation of data, Drafting or revising the article, Contributed unpublished essential data or reagents; JST, Conception and design, Acquisition of data, Contributed unpublished essential data or reagents; SN, HI, Analysis and interpretation of data, Drafting or revising the article; FC, Conception and design, Contributed unpublished essential data or reagents; TK, NM, Acquisition of data, Contributed unpublished essential data or reagents; KA, Conception and design, Analysis and interpretation of data, Drafting or revising the article; YK, Conception and design, Drafting or revising the article, Contributed unpublished essential data or reagents; KS, Conception and design, Acquisition of data, Analysis and interpretation of data, Drafting or revising the article, Contributed unpublished essential data or reagents

### Author ORCIDs

Fabrizio Mammano, http://orcid.org/0000-0002-9193-7696

## Additional files

### Supplementary files

• Supplementary file 1. Technical details of MCMC computations.

• Supplementary file 2. Estimated parameter values for Jurkat cell growth.

• Supplementary file 3. Estimated initial values for HIV-1 infection.

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
