## [Decision Letter]

Thank you for submitting your work entitled “Cell-to-cell infection by HIV contributes over half of virus infection” for peer review at *eLife*. Your submission has been favorably evaluated by Naama Barkai (Senior Editor), a Reviewing Editor, and two reviewers. One of the two reviewers, Vitaly Ganusov, has agreed to share his identity.

The reviewers have discussed the reviews with one another and the Reviewing Editor has drafted this decision to help you prepare a revised submission.

While others have previously shown that cell-to-cell spread is a rapid and efficient form of infection, the novelty of this work is in its attempt to quantify the contributions of cell-to-cell spread and cell-free infection in the same culture. However, there are three major concerns about the connections between the experiments and assumptions made in the modeling that need to be addressed in order to ascertain that the conclusions are correct. These major concerns are:

1) The parameter, *ω*, which measures the infection rate by cell-to-cell spread is obtained by fitting the model to data, and it is assumed that *ω*=0 in the experiments with shaking. *β*, the cell-free infection rate, is assumed to be the same under both infection conditions. In the data, shaking reduces infection, consistent with decreased cell-to-cell spread. However, it is also consistent with several other possibilities. For example, reduction of both cell-to-cell spread and cell-free infection due to many possible factors including lower production rates or infectivity of virus. If this is true, a similar attenuation of viral dynamics would be observed upon shaking but for a completely different reason, and invalidate the assumptions made to fit the data. To prove that the attenuated viral dynamics are strictly the result of a loss of cell-to-cell spread, a control experiment is needed. It is up to you to choose the appropriate control experiment that establishes this. We think that the following experiment might be appropriate:

Produce cell-free virus by filtering out any cells, then add it to either shaking or static Jurkat cell cultures and measure the percent infected cells under each condition at a few time points before two days (i.e. before the second cycle of infection). If shaking only affects cell-to-cell transmission, the results should be identical for shaken or static cultures in this experiment.

2) The parameter values are taken to be constant in time, but this may not be the case. Consider the parameter *g*, as an example. Cells that reach a maximum density may go into growth arrest and there is a lag time to get them out of it. This would mean that g is not constant. Is there experimental evidence that the parameters do not change with time. Also, at high levels of infection, which appears to be true in the experiments, is the cell culture still stable? If not, *g* would not be a constant.

3) It seems that the simulations are carried out without virus removal (without c) for 24 hours, and then an instantaneous removal occurs at 90% (not continuous removal of the virus as 2.3 per day). This set-up does not seem congruent with the experiments. The simulations should be carried out in a similar way to the experimental set-up to make sure that the parameter values are properly estimated. In addition, is it certain that cells/infected cells are not removed in the experiments, as it seems is assumed in the model? Finally, more quantitative estimates of agreement between the fitted models and experimental data are required. For example, by visual inspection of Figure 2, one could argue that the model does not describe the loss of infected cells and the virus after the peak well (Figure 2), and in Figure 2, it is not clear how good the fits are. Error bars need to be provided for the estimated parameters in Table 1.

[Editors' note: further revisions were requested prior to acceptance, as described below.]

Thank you for resubmitting your work entitled “Cell-to-cell infection by HIV contributes over half of virus infection” for further consideration at *eLife*. Your revised article has been favorably evaluated by Naama Barkai (Senior Editor), a Reviewing Editor, and one of the original reviewers. The manuscript has been improved but there are some remaining issues that need to be addressed before acceptance, as outlined below:

We have read your response and revised manuscript, and believe that you have not addressed the first major and essential point. As discussed in the first report, the assumption you make is that shaking eliminates cell-to-cell spread but does not affect cell-free infection. The premise of the whole paper is based on this assumption. According to the data presented in Figure 2, this may not be the case: Day 0 infection as shown in the middle panels of Figure 2 (static) and Figure 2 (shaking) is strictly the result of cell-free infection, as you indicate in the Methods. In Figure 2 (static), the number of cell infected is between 10^3^ to 10^4^/ml. In Figure 2 (shaking), the number of infected cells – by what is supposed to be the same cell-free infection input according to the methods – seems to be undetectable at day 0, with the first measurements appearing at a later time. This indicates that shaking, at least as performed by the method used in this paper, decreases cell-free infection. We requested that you perform a control experiment to show that this is not the case, and that shaking does not affect cell-free infection. We had also suggested a simple experiment for you to consider: add cell-free virus to static and shaking cultures and show that the infection rate is unaffected by shaking at the first infection cycle (subsequent cycles would involve cell-to-cell spread due to the co-culture, and would not be relevant). Rather than carry out this (or some other) control experiment, you point to a past paper that showed this. But, given the data in Figure 2 (see above), it is unclear that the assumption that shaking does not affect cell free infection is correct in your experiment. Since the rest of the work presented in the paper relies on this assumption, a control experiment demonstrating this assumption to be true is necessary.

---

## [Author Response]

*While others have previously shown that cell-to-cell spread is a rapid and efficient form of infection, the novelty of this work is in its attempt to quantify the contributions of cell-to-cell spread and cell-free infection in the same culture. However, there are three major concerns about the connections between the experiments and assumptions made in the modeling that need to be addressed in order to ascertain that the conclusions are correct. These major concerns are*:

*1) The parameter,* ω*, which measures the infection rate by cell-to-cell spread is obtained by fitting the model to data, and it is assumed that* ω*=0 in the experiments with shaking.* β*, the cell-free infection rate, is assumed to be the same under both infection conditions. In the data, shaking reduces infection, consistent with decreased cell-to-cell spread. However, it is also consistent with several other possibilities. For example, reduction of both cell-to-cell spread and cell-free infection due to many possible factors including lower production rates or infectivity of virus. If this is true, a similar attenuation of viral dynamics would be observed upon shaking but for a completely different reason, and invalidate the assumptions made to fit the data. To prove that the attenuated viral dynamics are strictly the result of a loss of cell-to-cell spread, a control experiment is needed. It is up to you to choose the appropriate control experiment that establishes this. We think that the following experiment might be appropriate*:

*Produce cell-free virus by filtering out any cells, then add it to either shaking or static Jurkat cell cultures and measure the percent infected cells under each condition at a few time points before two days (i.e. before the second cycle of infection). If shaking only affects cell-to-cell transmission, the results should be identical for shaken or static cultures in this experiment*.

In the original publication (Sourisseau et al., J Virol, 2007) describing the experimental model that we used here, the authors carefully verified that shaking did not induce nonspecific consequences on HIV infection. Namely, they verified that static and shaking conditions did not induce differences in cell viability, cell growth rate, expression level of receptor/co-receptors and adhesion molecules (Figure 2 in Sourisseau et al., J Virol, 2007). The authors also verified the absence of effect on virus release and on the efficiency of cell-free virus infection (Figure 3 in Sourisseau et al., J Virol, 2007). The previous demonstration of the absence of nonspecific consequences on HIV replication is now indicated in the manuscript (subsection “Adaptation of a mathematical model to explicitly consider cell-free and cell-to-cell infection”).

*2) The parameter values are taken to be constant in time, but this may not be the case. Consider the parameter* g*, as an example. Cells that reach a maximum density may go into growth arrest and there is a lag time to get them out of it. This would mean that g is not constant. Is there experimental evidence that the parameters do not change with time. Also, at high levels of infection, which appears to be true in the experiments, is the cell culture still stable? If not,* g *would not be a constant*.

Thank you for pointing this out, we would like to explain our assumption in detail, especially for the growth rate. In [Disp-formula equ1], we assumed the growth of target cells is described by *g*{1−(*T*(*t*)+*I*(*t*))/*T*_*max*_}*T*(*t*). This is called “logistic growth”. In this formulation, *g*{1−(*T*(*t*)+*I*(*t*))/*T*_*max*_} represents the average growth rate per cell. That is, if the density of total cells (i.e. *T*(*t*)+*I*(*t*)) is low, then the growth rate is approximately *g*, but if the density is high (i.e., around maximum density, *T*_*max*_), then the growth rate becomes 0, which models cell growth arrest. Because the term of 1−(*T*(*t*)+*I*(*t*))/*T*_*max*_ is a function of time, the average growth rate gradually decrease from 1 to 0 as the density increases from low to maximum, which shows a lag time. Thus, the logistic formulation well captures the density-dependent cell growth in cell culture as we previously showed in [6]. In contrast, for other parameters, we simply assumed they are constant. However, at least, in cell culture system, many papers including our own studies ([6]; Iwami et al., Retrovirology, 2012; Iwami et al., Front Microbiol., 2012; Komarova et al., 2013; [10]; Beauchemin et al., 2008) has shown that the simple constant assumption sufficiently reproduced the infection dynamics and quantified parameters reasonably. Therefore, we would like to keep our parameters constant in order to avoid the complexity of time-dependent parameters and to allow the use of our novel model for the cell-to-cell and cell-free HIV-1 infections.

*3) It seems that the simulations are carried out without virus removal (without c) for 24 hours, and then an instantaneous removal occurs at 90% (not continuous removal of the virus as 2.3 per day). This set-up does not seem congruent with the experiments. The simulations should be carried out in a similar way to the experimental set-up to make sure that the parameter values are properly estimated. In addition, is it certain that cells/infected cells are not removed in the experiments, as it seems is assumed in the model? Finally, more quantitative estimates of agreement between the fitted models and experimental data are required. For example, by visual inspection of*
Figure 2*, one could argue that the model does not describe the loss of infected cells and the virus after the peak well (*Figure 2*), and in*
Figure 2*, it is not clear how good the fits are. Error bars need to be provided for the estimated parameters in*
Table 1*.*

In addressing this very relevant and important comment, we have made a number of changes to our manuscript but also to our analysis itself. First, we would like to explain our assumption in [Disp-formula equ1]-[Disp-formula equ3] for the removal due to the experimental samplings. For each of the daily measurements of the virus concentration, the medium in our experiments was harvested, reducing the viral concentration by 90%. This removal can be captured using [Disp-formula equ3] by approximating the punctual removal of virus at each sampling time as a continuous, exponential decay of the viral load over the period between samples. In such a case, parameter *c* corresponds to the sum of the rate of virus loss due to harvesting of the medium plus the rate of loss due to degradation of the extracellular virus (which is negligibly small). In addition, on a daily basis, 10% of the cells in the culture were harvested to measure the number of target cells and infected cells. Similarly, the removals of the target and infected cells were included in the value of *g* and δ, respectively. We clarified this assumption by adding a few sentences to the subsection “Adaptation of a mathematical model to explicitly consider cell-free and cell-to-cell infection”.

Author response image 1.Punctual model for parameter estimation.The time course of experimental data for the numbers of uninfected cells (top) and infected cells (middle), and the amount of viral p24 antigen (bottom) in the static (left) and shaking (right) culture systems, respectively. The solid curves depict the best fit of the punctual model to the time-course dataset. All data in the experiment 3 were fitted simultaneously. For the experiments 1 and 2, we obtained similar fitting and parameter estimation values (data not shown).**DOI:**
http://dx.doi.org/10.7554/eLife.08150.017

As an example, we have expanded our analysis using [Disp-formula equ1]-[Disp-formula equ3] to also include an analysis of the experimental data using the model with punctual removal, i.e. we determined best-fit parameters for this punctual model and present graphs of its agreement to both data sets over time (Figure 5). Generally, the punctual removal model describes the experimental data well, and the estimated values for the model parameters are similar to those found using [Disp-formula equ1]-[Disp-formula equ3]; *β*=3.49×10^−6^, *ω*=1.17×10^−6^, *p*=0.53, *δ*=0.50 in [Disp-formula equ1]-[Disp-formula equ3], and *β*=6.65×10^−6^, *ω*=1.69×10^−6^, *p*=0.13, *δ*=0.43 in the punctual model. In our earlier works ([6]; Iwami et al., Retrovirology, 2012; Iwami et al., Front Microbiol., 2012; [10]) we have also shown that approximating punctual removal as a continuous exponential decay has minimal impact on the model parameters and provides an appropriate fit to the experimental data. And also, unfortunately, we could not define the basic reproduction numbers such as *R*_*cf*_, *R*_*cc*_, and *R*_0_ for the punctual model because of its discontinuity. Quantifying these values and comparing them are key findings of this paper. Therefore, we used the exponential decay approximation (i.e. [Disp-formula equ1]-[Disp-formula equ3] here. Nevertheless, we do feel that inclusion of the above discussion about the punctual removal is a valuable addition to our manuscript and improves the completeness of our work (see the second paragraph of the subsection “Adaptation of a mathematical model to explicitly consider cell-free and cell-to-cell infection”).

Furthermore, for more quantitative estimates of agreement between the fitted models and experimental data, we performed Bayesian estimation for the whole dataset using Markov Chain Monte Carlo (MCMC) sampling. The Bayesian estimation enables us to assess the variability of kinetic parameters and model prediction as posterior predictive intervals (see Results, Methods and [Supplementary-material SD1-data] We thank the reviewers for leading us in this direction.

[Editors' note: further revisions were requested prior to acceptance, as described below.]

*We have read your response and revised manuscript, and believe that you have not addressed the first major and essential point. As discussed in the first report, the assumption you make is that shaking eliminates cell-to-cell spread but does not affect cell-free infection. The premise of the whole paper is based on this assumption. According to the data presented in*
Figure 2*, this may not be the case: Day 0 infection as shown in the middle panels of*
Figure 2
*(static) and*
Figure 2
*(shaking) is strictly the result of cell-free infection, as you indicate in the Methods. In*
Figure 2
*(static), the number of cell infected is between 10*^*3*^
*to 10*^*4*^*/ml. In*
Figure 2
*(shaking), the number of infected cells – by what is supposed to be the same cell-free infection input according to the methods – seems to be undetectable at day 0, with the first measurements appearing at a later time. This indicates that shaking, at least as performed by the method used in this paper, decreases cell-free infection. We requested that you perform a control experiment to show that this is not the case, and that shaking does not affect cell-free infection. We had also suggested a simple experiment for you to consider: add cell-free virus to static and shaking cultures and show that the infection rate is unaffected by shaking at the first infection cycle (subsequent cycles would involve cell-to-cell spread due to the co-culture, and would not be relevant). Rather than carry out this (or some other) control experiment, you point to a past paper that showed this. But, given the data in*
Figure 2
*(see above), it is unclear that the assumption that shaking does not affect cell free infection is correct in your experiment. Since the rest of the work presented in the paper relies on this assumption, a control experiment demonstrating this assumption to be true is necessary*.

According to the suggestion raised by the editors (“it is unclear that the assumption that shaking does not affect cell free infection is correct in your experiment”), we carried out an additional experiment and the data is shown as Figure 2—figure supplement 1. As shown, we have clearly verified that the shaking procedure does not affect the efficacy of cell-free infection.